

# Deformation mechanisms and evolution of the microstructure of gouge in the Main Fault in Opalinus Clay in the Mont Terri rock laboratory (CH)

Ben Laurich[1,2], Janos L. Urai[1], Christian Vollmer[3], Christophe Nussbaum[4]

[1]Institute for Structural Geology, Tectonics and Geomechanics, RWTH Aachen University, Lochnerstrasse 4-20, D 52056 Aachen, Germany
[2]Now at: Federal Institute for Geosciences and Natural Resources (BGR), Stilleweg 2, D-30655 Hannover, Germany
[3]Institute for Mineralogy, University Münster, Corrensstraße 24, D-48149 Münster, Germany
[4]Swiss Geological Survey, Federal Office of Topography Swisstopo, Seftigenstrasse 264, CH -3084 Wabern, Switzerland

*Correspondence to*: Ben Laurich (ben.laurich@bgr.de)

**Abstract.** We studied gouge from an upper-crustal, low offset reverse fault in slightly overconsolidated claystone in the Mont Terri rock laboratory (CH). The laboratory is designed to evaluate the suitability of the Opalinus Clay formation (OPA) to host a repository for radioactive waste.

The macroscopically dark gouge displays a matrix-based, P-foliated microfabric bordered and truncated by µm-thin shear zones consisting of aligned clay grains, as shown by BIB-SEM and optical microscopy. TEM-SAED shows evidence for randomly oriented nm-sized clay particles in the gouge matrix, surrounding larger elongated phyllosilicates with a strict P-foliation. For the first time in OPA, we report the occurrence of amorphous $SiO_2$ grains within the gouge. Gouge has lower SEM-visible porosity and almost no calcite grains, compared to undeformed OPA.

We present two hypotheses to explain the origin of gouge in the Main Fault: (i) "authigenic generation": fluid-mediated removal of calcite from deforming OPA during shearing, (ii) and "clay smear": mechanical smearing of calcite-poor (yet to be identified) source layers into the fault zone. Based on our data we prefer the first or a combination of both, but more work is needed to resolve this.

Microstructures indicate a range of deformation mechanisms including solution-precipitation processes and a gouge which is weaker than OPA because of the lower fraction of hard grains. We infer that the long-term rheology of gouge is more strongly rate-dependent than suggested from laboratory experiments.

## 1 Introduction

Gouge is a fine-grained fault rock common in near-surface faults [*Sibson*, 1977; *Vrolijk and van der Pluijm*, 1999]. It influences the mechanical and hydraulic properties of a fault, can act as a fluid-barrier or conduit and typically forms a zone of mechanical weakness, which localizes further faulting. Hence, understanding the gouges' physical rock properties is essential in hydrocarbon exploration and production, hydrogeology, earthquake research and nuclear waste disposal.



As a follow-up to our publications on µm-thin shear zones and on scaly clay from the *Main Fault* in the Mont Terri underground research laboratory (MT URL) [*Laurich et al.*, 2014, 2017; *Laurich*, 2015], we address in this paper the mineralogy and microstructure of gouge from the *Main Fault* to infer physical rock properties, underlying deformation mechanisms and the evolution of the gouge.

In this study, we build upon many gouge-related microstructural studies. Although we cannot cite all of these, we include a categorized literature overview in the appendix (without a claim for completeness).

### 1.1 Geology and sample origin

As its main objective, the MT URL evaluates the long-term safety of radioactive waste disposal. A detailed description on the local geological setting can be found in Nussbaum et al. [2011, 2017], in Jaeggi et al. [2017], in Bossart and Thury [2008],

and in Becker [2000]. The *Main Fault* is a low offset (<80 m) reverse fault visible as a 0.8 - 3 m wide fault zone in the shaly facies of the Opalinus Clay formation (OPA) at the MT URL. The fault zone is heterogeneous: It consists of gouge, scaly aggregates [sensu *Vannucchi et al.*, 2003], S-C bands, meso-scale folds, microfolds, numerous micron-thick shear zones, which form shiny slickensided surfaces when broken and blocks of undeformed OPA (Figure 1).

The environmental and lithological controls at the onset of faulting in the late Miocene [*Nussbaum et al.*, 2011, 2017] have

been inferred as follows: 55 °C temperature; 1000 m overburden; sub-horizontal, NNW-SSE oriented $\sigma_1$; progressive faulting in a shear-fault bend fold (the MT anticline); calcite and celestite veins indicating paleo-fluid flux during faulting; slightly over-consolidated protolith (max. burial 1350m, 85 °C) with 8-24% porosity [*Mazurek et al.*, 2006; *Nussbaum and Bossart*, 2008; *Houben et al.*, 2013; *Haller et al.*, 2014; *Laurich et al.*, 2014]. The present permeability of OPA is very low with no significant hydrological contrast between wall rock and the Main Fault (2 x 10$^{-13}$ m/s, Nussbaum and Bossart, 2008). Profiles

of a range of pore-fluid geochemical tracers are not perturbed near or within the Main Fault [*Mazurek et al.*, 2011]. OPA has a strongly anisotropic unconfined compressive strength between 6 and 28 MPa [*Bock and Blümling*, 2001; *Amann et al.*, 2011]. Gouge is present in all outcrops of the *Main Fault* known to the authors. It occurs as a thin (up to 2 cm), but continuous dark band at the upper fault zone boundary in Figure 1 but also occurs inside the *Main Fault* zone. Gouge can be recognized by is its dark, matte appearance compared to undeformed OPA. Gouge has sharp boundaries and is at least to one side always

bordered by scaly clay.

The aim of this paper is to characterize gouge microstructure by optical microscopy and BIB-SEM and FIB-TEM methods, to evaluate the evolution of gouge during movement of the Main Fault, and finally to discuss the mechanical and fluid flow properties of the Main Fault during its geological evolution.

### 2 Methods

The samples were retrieved from drillcores BIC-A1, BPS-12 and from all Main Fault outcrops in the MT URL. Drill core samples have the well name as a prefix, outcrop samples are named with the prefix 'V' or 'A'. The samples were transported



in vacuum-sealed bags and resin-stabilized in the lab. Due to the fragile nature of the specimen, sampling and sample preparation was conducted with great care and without the use of water (except when on purpose and explicitly stated).

## 2.1 Microstructure

The resin stabilized samples were sectioned approximately parallel to the outcrop wall, i.e. parallel to the inferred main

principle displacement and perpendicular to bedding. The cut samples were further treated in three different ways: (1) by water immersion, (2) by ultra-thin sectioning and (3) by sub-sampling for broad-ion-beam (BIB) polishing. Some polished samples were decorated by spraying a thin water film on the polished sample surface using an aerosol can. Immediate drying of the water results in a fine decoration of the foliation by differential swelling of the clay matrix. Ultra-thin (<10µm) sections for optical microscopy were produced by Geoprep (Basel). Sub-samples were cut with a thin (0.3 mm) circular diamond saw,

manually ground using abrasive paper down to 2400 grit and subsequently BIB-polished. Two Argon-BIB devices were used: (1) a JEOL SM-09010 operating at 6 kV, 150 – 200 µA for 7.5 – 10h, producing 2 mm² sections and (2) a Leica TIC3X operating at 5 - 7.5 kV, 2 – 2.8 mA for 2 h, with angle varying between 4.5° - 10.5°, polishing sub-sample surfaces of up to 1 cm².

The samples were imaged using an optical as well as a scanning electron microscope (SEM): A Zeiss Supra55 field emission

SEM operating at 3 - 30 kV in secondary electron (SE) and in back scattered electron (BSE) mode and also using energy-dispersive X-ray spectroscopy (EDX). Water decorated samples were imaged in oblique incident light with optical microscopy. Ultra-thin sections were imaged in transmitted light microscopy (xpol, full waveplate) and in the SEM. A total of 11,124 grains were manually segmented from BIB-SEM BSE micrographs. Segmentation and statistics were done using ArcGIS 10.3 [*ESRI*, 2014], ImageJ 1.51a [*Schindelin et al.*, 2015], RStudio 0.99 [*RStudio Team*, 2015] and MS Excel 2013.

BIB-SEM and corresponding statistical procedures are described in detail in other microstructural clay studies from the Aachen GED Group [*Houben*, 2013; *Hemes*, 2015; *Klaver*, 2015; *Laurich*, 2015; *Desbois et al.*, 2016].

Transmission electron microscopy (TEM) techniques were applied on focused ion beam (FIB) lamellae of 10 x 5 x 0.15 $\mu m^2$ (length, width, thickness). We used a FEI Strata 205 FIB (Central Facility for Electron Microscopy, University of Aachen, tungsten coating, 30 kV, 20 nA - 100 pA) and a Zeiss Libra 200FE (200 kV Schottky field emitter, Koehler illumination

system, in-column Omega energy filter, Institute for Mineralogy, University of Münster). High angle annular dark field (HAADF) scanning TEM (STEM) and bright field (BF) images were collected, giving information on average atomic number and mainly diffraction contrast, i.e. crystallinity of illuminated areas, respectively. We performed EDX measurements to specify major element characteristics using a Si(Li) detector. To minimize sample damage, only short dwell times were applied. Selected area electron diffraction (SAED) patterns were recorded to specify crystallographic parameters of regions of interest

within the FIB lamella.



## 2.2 Bulk analyses

In addition to electron microscopy-based EDX, we used X-ray diffraction (XRD) to determine mineralogical phases. XRD was performed on two different devices: A Bruker D5000 and a Huber MC9300. Quantification was done by Rietveld refinement, using the software TOPAS and BGMN, for patterns recorded with the Bruker D5000 and the Huber MC9300, respectively. We followed procedures as published in Kahle et al. [2002] and Ufer et al. [2008]. Diffractograms were recorded with 2Θ ranging from 2° to 92°, with long counting times. EDX was accomplished by silicon drift detectors at both electron microscopy devices: SEM and TEM.

Vitrinite reflectance (VR) and total organic carbon (TOC) of gouge and surrounding wall rock were determined at the Institute of Geology and Geochemistry of Petroleum and Coal, RWTH-Aachen. VR was measured using a Zeiss Axio Imager microscope. The sample preparation was done perpendicular to bedding foliation by dry polishing with carbide abrasive papers down to 4000 grit. We followed the procedure in Littke et al. [2012] and Bou Daher et al. [2014]. TOC measurements were performed with a liquiTOC II analyzer. The temperature ramp was 300 °C/min up to 550 °C, released $CO_2$ was detected with a non-dispersive infrared detector.

## 3 Results

### 3.1 Microstructure

Gouge displays a beautiful geometric microstructure. In the outcrop, it stands out due to its matte black color in surfaces broken perpendicular to the foliation, and wavy, smooth, shiny slickensided shear surfaces (Figure 2 - Figure 9). Gouge can be disintegrated into fine powder between two fingers and becomes sticky when wet. It displays sharp boundaries to the surrounding wall rock, often in a distinct straight line (e.g. Figure 3d and e), but sometimes highly irregular (Figure 9). The wall rock is either undeformed protolith or scaly clay, and often enriched in micro-calcite veins that are frequently strained and in Riedel orientation to the main shortening axis of the Main Fault (Figure 3a, b, c and e, wall rock veins in Figure 9c). There are no veins inside the gouge. Figure 17 displays a generic sketch of all microstructural elements in gouge.

### 3.1.1 Gouge-internal foliation and gouge-internal shear bands

The gouge-internal foliation can be decorated by the water immersion technique (Figure 3, Figure 4). It shows a distinctive, well-organized P-foliation [orientation nomenclature sensu *Logan et al.*, 1979] with a high fabric intensity [sensu *Yan*, 2001; *Haines et al.*, 2009]. Correspondingly, ultra-thin sections of gouge display a uniform extinction pattern along the P-foliation under crossed polars [not shown here, refer to *Laurich et al.*, 2014].

By slight foliation and brightness variations, two types of gouge can be differentiated: (1) less dark gouge with a high angle P-foliation to shear direction and (2) dark gouge with a low angle or almost parallel P-foliation to shear direction (Figure 3, Figure 4, Figure 9). By SEM-EDX, the brightness differences between both gouge types can be attributed to Ca content, which





is lowest in type 2, followed by type 1 and highest in the wall rock and in wall rock clasts (Figure 3b+c, Figure 4). The border of both gouge types is usually very sharp along Y-orientation (Figure 3e, Figure 4b). Gouge type 2 is occasionally bordered by type 1 on both sides (Figure 4), but has in general one border to brighter wall rock (Figure 3d).

The gouge-internal foliation is reflected in a slight sigmoidal 'drag-in' shape at the gouge zone boundaries (Figure 3d and e,

Figure 8), where µm-wide bands of shear-parallel foliation resemble Y-shears. Such bands of parallel foliation can be spotted inside the gouge, too. They occur mostly in Y- and R- and less abundant in R'-orientation (exemplarily outlined in Figure 4). Framed between these bands, the P-foliation in gouge type 1 displays a sigmoidal style, comprising an S/C fabric (Figure 3a+b, left in type 1 gouge and Figure 4). The foliation pattern of gouge type 1 continues into the scaly wall rock (right part in Figure 3a, b). The gouge-internal bands can also be seen in large-area BIB polished samples imaged by reflected light microscopy in

Figure 7c and d. The Figure displays a wavy net of approx. 30 µm wide bands that are slightly darker than the gouge matrix. In the SEM, the dark bands were identified as semi-continuous, 15 µm to 50 µm wide shear bands, which expose by a distinct change in particle orientation (Figure 10 - Figure 14). The borders of the gouge-internal shear bands can be very sharp (Figure 13a+b, Figure 14d) or diffuse, such as the lower shear band boundary in Figure 10b. To define such a diffuse boundary is subjective and other interpreters might trace a number of branches of the shear band, which often nucleate and die out next to

larger grains. The level of detail here is in the order of the grain size. In a similarly subjective fashion, we traced a number of µm-wide shear zones (Figure 14b) that run through the shear bands and that define the bands' sharp borders as Y-shears (Figure 14d), consistent with findings by Logan et al. [1992].

Contrary to undeformed OPA, the gouge microstructure contains no fossils, has a much higher fabric intensity and drastically less calcite grains. Many smaller open fractures are present, which we confidently interpret as artifacts from desiccation and /

or unloading [cf. *Dehandschutter et al.*, 2005; *Houben*, 2013]. Figure 10 provides a micrograph comparison of gouge and undeformed OPA (shaly facies).

### 3.1.2 Quantitative grain analysis

The following compares grain size distributions (GSD), orientations and shapes of undeformed OPA (sample V08) and gouge (sample V15-4). Both BIB-polished samples derive from the Main Fault: V08 from an undeformed part in gallery 08 (outcrop

to the SSW), V15-4 from the upper fault zone boundary in gallery 98 (outcrop to the SSW). The area examined by DIA of sample V08 is 7395 µm², with 6410 manually segmented grains that fall completely inside the area. The area of sample V15-4 is 3600 µm², with 5000 grains that fall completely inside the area (Figure 10).

Figure 11a displays the density GSDs of both samples following the power-law equation:

$$\frac{N_i}{(bw_i * At)} = C * bc_i^{-D} ,$$

eq. 1

where $N_i$ is the number of grains within bin $i$, having an exponential increasing bin width $bw_i$. The frequencies are normalized

by the examined area $At$ for sample comparability. The x-axis in Figure 11a displays the bin centers ($bc_i$), and $C$ is a constant

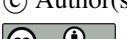



of proportionality. Both samples show an almost identical power-law exponent *D*, which has been derived by the least squares method using the logarithm of the data. The validity of both power-law models within the chosen ranges (boxes in Figure 11a) is supported by constant values in local slope [cf. *Bonnet et al.*, 2001 and references therein], shown as thin grey lines in Figure 11a. Therein, the local slope ($D_l$) is calculated by:

$$D_l = \frac{\log\left(\frac{Nn_i}{C}\right)}{\log(bc_i)} \ .$$

eq. 2

Deviations of recorded frequencies from the power-law (diamonds in Figure 11a) can be attributed to resolution limitations (truncation), i.e. the smaller the grains, the more often they are missed in the segmentation [cf. *Bonnet et al.*, 2001].

The parity in the power-law exponents (1.98 and 1.97 for V15-4 and V08, respectively) suggests that both samples have a similar GSD. However, this cannot hold true for values below the regression range: Only 30.7 area-% of sample V15-4 were segmented (either as grains or pores), the remaining (from here on termed matrix) being not SEM-resolvable. Contrary,

47.1 area-% were segmented in sample V08, although both BSE micrographs were recorded with equal magnifications and brightness contrasts and were both manually segmented with the same good level of detail (we recommend to zoom-in Figure 10). The effect of this difference in total segmented area is illustrated in the cumulative GSDs (Figure 11b): A continuous trend in the cumulative GSD of sample V15-4 beyond the range depicted in Figure 11a down to 1 nm² grain size sums up to a total grain area of 70 %, leaving 30 area-% unexplained. We therefore hypothesize a GSD for gouge with a very high contribution

of smallest grains below the resolution limit (~ 10,000 nm²). The undeformed sample V08 shows that 100 area-% would be realized by following the cumulative GSD trend down to 16 nm² grain size. Note, that in Figure 11b the cumulative GSDs do not add up to 100 area-%. The lacking part (6.67 and 1.62 area-%, for samples V08 and V15-4, respectively) is the area segmented as pores, open fractures and those grains that fall not completely inside the examined area and are thus excluded from the GSD analysis.

The rose diagrams in Figure 11c show a clear grain orientation difference inside and outside the shear band. The undeformed OPA shows also a small scatter in grain orientation, which reflects the bedding foliation (Figure 11c). Note, that entries in the rose histogram are weighted by their axial ratio in order to enhance the main fabric orientation (eq. 3 and eq. 5), i.e. a grains' contribution to an orientation bin increases with increasing elongation. Grains that are as wide as long (axial ratio = 1) do not contribute at all. Shape factor distributions are shown in Figure 11d. The proportionality of axial ratio and circularity (eq. 4),

wherein the axial ratio is larger than the circularity, is indicative for angular shapes with little surface roughness, such as the box-shaped, elongated phyllosilicates. Grains within undeformed OPA display small values in circularity more often than grains in gouge, while the axial ratio is similar in both samples. This difference in circularity can be attributed to a higher surface roughness for grains in undeformed OPA. Used equations are:





$$axial\ ratio = \frac{width}{length}, \qquad \text{eq. 3}$$

$$circularity = \frac{4\pi * area}{perimeter^2}, \qquad \text{eq. 4}$$

$$N = \sum_{1}^{k} m_i * (1 - axial\ ratio_i), \qquad \text{eq. 5}$$

where $N$ is the basic population, $k$ the number of bins ($k$ = 18 bins á 10°) and $m_i$ a function that counts the number of observations that fall into each bin. The raw data as well as their processing are provided as supplement to this article.

Analysis of porosity in the gouge is somewhat subjective because of the presence of open microfractures in the BIB-polished sample, which we interpreted as artefacts. Such fractures are much less common in undeformed OPA [*Houben et al.*, 2013] perhaps because gouge has a much stronger foliation. Still, the SEM-visible porosity in gouge including the fractures (Figure 10, Figure 12) is much smaller than in undeformed OPA (0.84 and 2.4 %, respectively). Gouge material in between larger fractures shows clearly less visible porosity than undeformed OPA at the same resolution (Figure 12). Undeformed OPA has

18 % porosity as best estimate [*Nussbaum and Bossart*, 2008; *Houben et al.*, 2014]. In recent work, high porosities for the OPA gouge (> 20 %) are stated by mercury injection porosimetry and He-pycnometry in Orellana et al. [2016], dominated by pores < 14 nm, however it is unclear if these measurements excluded artificial openings.

### 3.1.3 Gouge clasts

Gouge type 1 frequently incorporates bright wall rock clasts, which are oriented parallel to the P-foliation or in Y-orientation

(Figure 3, Figure 4, Figure 7, Figure 8 and Figure 9). Often they match shape and orientation of the adjacent wall rock boundary and we interpret them as detached wall rock material (Figure 8, Figure 9). Nevertheless, from the pictures it remains unclear if they are teared-off from the wall by gouge intrusion or if their neighboring material got reworked into gouge or both. Wall rock clasts are frequently sheared along gouge internal Y- and R-shears. Arrays of wall rock clasts typically resemble the appearance of one half of a feather: Their sigmoidal clasts are P-foliation parallel aligned such as several feather-spikes that

are attached to a Y-shear being the feather-shaft (Figure 8a, c).

Contrary to wall rock clasts, mineral grain clasts [or "survivor grains", *Cladouhos*, 1999b] are rare in gouge type 1 and frequent in gouge type 2 (Figure 3, Figure 4). They are large, P-foliation parallel particles of calcite or feldspar, which are often fractured and show spalled fragments. Typically these fragments are located in pressure shadows of the grains (larger grain in the very left of Figure 4a).

### 25   3.1.4 Gouge matrix

BIB-SEM (BSE) shows grains down to about 100 nm width (Figure 13). Yet, these grains are clasts to a matrix-based fabric of even smaller particles, which are not resolvable, even by high resolution TEM, because the FIB lamella thickness is clearly larger than individual particle sizes (Figure 15b). TEM-EDX and SAED indicates that the matrix comprises ultra-fine grained



(< 10 nm), polycrystalline clay particles in various orientations (Figure 15c, cf. Viti [2011]). The presence of such small particles is in agreement with the GSD above, which states an amount of small grains positively deviating from the power-law trend of larger grains. The matrix (respectively unsegmented region) makes up 69.3 area-% of the gouge sample (for comparison: 52.9 area-% are unsegmented in the undeformed sample, see Figure 10).

## 3.2 Bulk mineral analysis (XRD, TOC and VR)

A total of 18 XRD measurements were performed: 8 on powdered gouge material and 10 on surrounding wall rock material. All samples are equal in type and quantity of minerals, except for a reduced amount of calcite in gouge. There is no spatial trend in calcite or any other mineral composition towards gouge, as evident from congruent XRD patterns in Figure 6. The boundary between bright wall rock and dark, calcite reduced gouge is sharp. This is in agreement to our SEM-EDX observations (e.g. Figure 3c).

The dark color of gouge is not a result of a high TOC: one gouge and one wall rock sample show similar TOC values (1.1 wt.-% and 1.3 wt.-%, respectively). These values are consistent with literature on OPA [*Nagra*, 2002; *Pearson et al.*, 2003].

Frictional heating in the gouge is not evident from VR. One dry polished sample containing gouge and wall rock shows values ranging from VR = 0.55 % - 1.2 %, with an average of 0.948 % and 0.915 % for the gouge (N = 40) and the wall rock part (N = 70), respectively. These values are consistent with literature of OPA at the Mont Terri location [*Nagra*, 2002].

## 3.3 Indicators of deformation mechanisms

We identified microstructural indicators for different mechanisms: (1) cataclasis and abrasion, (2) pressure solution precipitation, (3) neoformation of clay grains, (4) intracrystalline plasticity and (5) frictional granular flow.

### 3.3.1 Cataclasis and abrasion

The shape factor analysis above points to a process which produces rounded grains with smooth boundaries, presumably by an abrasion process perhaps coupled with dissolution (e.g. Figure 10 and slight shift in circularity shown in Figure 11d). Moreover, trans-granular fracturing and boudinage is abundant in mica, calcite, feldspar and $SiO_2$ grains across several scales (Figure 13). Figure 16 shows bending and breakage of a mica grain by delamination along the (001) basal planes with progressive break-up even of smallest grains. Particle breakage can be accomplished along kinks as well (Figure 15d). Commonly, fractured grain parts are offset so far that they cannot be correlated anymore and the full extent of grain fracturing cannot be quantified [*Milliken and Reed*, 2010; cf. *Haines et al.*, 2013].

### 3.3.2 Diffusive mass transport (pressure solution precipitation and clay neoformation)

In the sections above, we reported strongly reduced calcite content for gouge, while in the wall rock calcite veins and calcite patches are abundant. If the alternative clay-smear explanation (discussed below) can be excluded, this is a critical discovery





that indicates large scale removal by pressure solution of calcite from the gouge and perhaps crystallization in the nearby microveins [*Laurich et al.*, 2014, 2017; *Clauer et al.*, 2017].

Isotope and REE analyses of gouge material show differences to undeformed OPA, to veins and to scaly clay aggregates from the Main Fault, suggesting that gouge interacted with fluids chemically different from those in the other parts of the Main Fault

[*Clauer et al.*, 2017].

Figure 5 shows a slickensided surface of a broken gouge sample. At high magnifications, the smoothly polished surface shows nm-sized clay particles. These nm-sized clay particles, also reported in the gouge matrix in TEM (Figure 15b), may result from neoformation, as described for gouge in other studies [*Vrolijk and van der Pluijm*, 1999; *Sasseville et al.*, 2008; *Schleicher et al.*, 2010; *Warr et al.*, 2014]. Particularly concerned with illite neoformation in the Main Fault, Clauer et al. [2017] report

preliminary K-Ar ages of nm-sized illites from the gouge material that coincide with the inferred onset of the Main Fault (9-4 Ma).

Above, we propose that the nm-sized clay grains could be detrital material and/or a result from cataclasis and/or neoformation. Additionally, they could derive from clay mineral transformation. This mechanism has been reported for the transformation of smectite and/or mixed-layer illite-smectite to illite in other fault gouges [*Vrolijk and van der Pluijm*, 1999; e.g. *Dellisanti et*

*al.*, 2008]. The shaly OPA facies contains 5-20 wt.-% mixed-layer illite-smectite [*Pearson et al.*, 2003]. Similar to the setting of the Main Fault, Casciello et al. [2011] report an illitization for an upper crustal, low-offset fault in a mudstone protolith. They argue that illitization can occur localized in shear bands at low P-T conditions and without fluid-rock interaction. Further, and similar to our observations, they report darkened shear zones and relate this to the localized illitization of the fault rock. However, we do not have sufficient evidence for such an illitization process in OPA, as we did not design our XRD

measurements to spot differences in smectite / illite contents and our SEM-EDX measurements yield no element contrast for the μm-thin bands.

The gouge comprises well rounded, anhedral, amorphous $SiO_2$ grains, shown by the diffuse SAED ring pattern of one $SiO_2$ grain shown in Figure 16a. We can exclude the possibility that amorphous $SiO_2$ is an artefact of the FIB preparation procedure as the less stable clay minerals in the lamella are preserved. Consequently, the amorphous $SiO_2$ may have formed by

precipitation from hydrothermal fluids. Amorphous $SiO_2$ has not been reported for OPA before [*Pearson et al.*, 2003; e.g. *Lerouge et al.*, 2014]. Gaucher et al. [2003] even concluded that amorphous silica was not found in OPA and that neoformed quartz cement formed during diagenesis is well preserved with uncorroded crystal planes. The mineral assemblage of amorphous material and clays is typical in several gouge localities although in more mature fault gouges [*Power and Tullis*, 1989; *Di Toro et al.*, 2004; *Hadizadeh et al.*, 2012; *Janssen et al.*, 2013; *Kirkpatrick et al.*, 2013; *Kameda et al.*, 2017].

### 3.3.3 Intracrystalline plasticity

Intracrystalline plasticity of phyllosilicates is proposed as a deformation mechanism in clay [*Urai and Wong*, 1994]. However, it is the most difficult one to observe: permanent deformation without fracturing is thought to occur mostly along the clays' (001) basal planes, producing a crystal lattice hardly distinguishable from the educt [*Warr and Cox*, 2001; *Warr et al.*, 2014].



Figure 16 shows grains delaminated along (001), suggesting intracrystalline plasticity [cf. *Goodwin and Wenk*, 1990]. Other indicators for intracrystalline plasticity are the frequently folded (e.g. Figure 14e) and bent (e.g. Figure 15a) clay particles, although elastic deformation may also cause this [*Wenk et al.*, 2008; *Kanitpanyacharoen et al.*, 2015]. In a nano-indentation test to determine the elasticity of single phyllosilicate minerals, Zhang et al. [2010] suggest measurement artefacts caused by intracrystalline plasticity beneath the indenter tip. Particle bending next to shears can be seen also in TEM: Figure 15 displays a 500 nm thick, dragged mica.

Moreover, $SiO_2$ grains in gouge are frequently boudinaged (e.g. Figure 13a), a result of strain likely induced by a plastic deformation of the surrounding gouge matrix (compare also with boudinaged mica in Figure 13c). $SiO_2$ grain cusps that form during this process are filled with clay minerals of the gouge matrix, while grains with trans-granular fractures often show unfilled pore space (e.g. $SiO_2$ grain in Figure 13c).

### 3.3.4 Frictional granular flow

The preserved high fabric intensity (Figure 3), the well-developed S/C fabric pattern (Figure 4) and the occasional 'mixing' and 'injection' appearance of gouge (Figure 9, cf. ultracataclasite in Chester et al. [1998]) suggest a continuous granular flow, where individual particles are aligned in a passive response to movement along shear bands and µm-thin shear zones. Therein, the individual particles pass-by each other by grain boundary sliding [*Passchier and Trouw*, 2005; *Haines et al.*, 2014], while the void between the grains is filled by the ultra-fine grained matrix (Figure 13, Figure 15) or, occasionally, by the development of pores in pressure shadows around the $SiO_2$ grains (Figure 14b, Figure 15d + e). We propose that the ultra-fine matrix was significantly weaker than the rest of the gouge [cf. *Jessell et al.*, 2009; *Han et al.*, 2011], possibly aided by water bound to the nm-grains. This mechanism led to the local strong alignment of nanoparticles, together with complete loss of preferred orientation of nanoparticles in other regions (Figure 15c). We also note that larger organic matter particles (OM) although strained, still appear stronger than the matrix (Figure 13) and could be used as strength gage [*Eseme et al.*, 2007].

### 3.4 Results summary

Gouge comprises a variety of microstructures (Figure 17). It occurs as a continuous, dark, fine-grained band at the upper boundary and in a few samples from within the scaly clay part of the Main Fault. It shows a reduction in calcite compared to the brighter wall rock and can be separated in two types. Type 1 shows high P-foliation angle to shear zone dip, bright wall rock clasts and S/C foliation. Type 2 shows lower P-foliation angle to shear zone dip, no wall rock clasts and several grain clasts. Both types have internal µm-thin shear zones and wider shear bands (<50 µm) mostly in the R-, P- and Y-orientations. We infer that offset was accommodated mostly along these shears, while gouge between responded in a passive, viscous reorientation of grains in an ultra-fine grained matrix acting as a solid lubricant. There are indicators for all major deformation mechanisms. Both gouge types can be classified as "foliated" [sensu *Cladouhos*, 1999a].



## 4 Discussion

We can envisage two very different mechanisms for the formation of the gouge in the Main Fault: (A) extreme reworking of OPA with removal of calcite during deformation, and (B) clay smear. A combination of these mechanisms is considerable, too. We will discuss both hypotheses in the following sections.

### 4.1 Clay smear from a source layer

It is possible that gouge developed (partly) by clay smear from a calcite-free source layer [sensu *Vrolijk et al.*, 2015]. Arguments against this hypothesis are: (i) such a source layer in the faulted stratigraphy (< 80 m) of the Main Fault has not been identified (but there has not been a focused search for it either), (ii) parity in mineral composition and (iii) in density GSD in gouge and undeformed OPA for grains > 10,000 nm² (Figure 11), as well as (iv) isotope signals that indicate fluid-driven mineral trans- and neoformations contemporaneous to the inferred tectonic activity [*Clauer et al.*, 2017]. Still, clay smear from a source layer could have acted, e.g. when several thin source beds were truncated and linked-up to yield sufficient gouge material. Dark, gouge-similar, bedding parallel bands are known from other localities in the MT URL (Figure 18). A fine smear-in from those with eventual fluid-rock interactions appears feasible.

### 4.2 Fluid-assisted reworking of the protolith

An alternative hypothesis is that the spatial relation of undeformed fabric, scaly clay and gouge type 1 and 2 is also a progressive relation: gouge was generated from highly deformed OPA, with gouge type 2 generated from type 1. This scenario, however, involves a complex interference of micromechanical processes, including external fluids, and it would be strongly supported by the sporadic occurrence of transition stage microstructures, which are not clearly recognizable. Below, we describe the processes in this scenario.

### 4.2.1 Stage 1 – development of scaly clay [cf. *Laurich et al.*, 2017]

In a first stage, undeformed protolith develops into a scaly clay fabric, with an increasing density of µm-thin, non-porous, shear zones and microlithons. The generation of scaly clay results from progressive development of new µm-thin shear zones in between relays, likely enhanced by a combination of two mechanisms: (1) geometric locking of existing µm-thin shear zones and (2) local fluid pressure increase, caused by the pore water expulsion from the non-porous shear zones [*Vannucchi et al.*, 2003; *Laurich et al.*, 2017]. The frequent veins along µm-thin shear zones are evidence of external fluid flow when the Main Fault was active [*Haller et al.*, 2014; *Clauer et al.*, 2017]. Therein, the µm-thin shear zones have been interpreted to have acted as temporary fluid conduits with non-porous, impermeable side walls. This assumption is supported by differing geochemical signatures ($^{87}Sr/^{86}Sr$) of veins to surrounding protolith and by water from borehole-inflow showing a $^{87}Sr/^{86}Sr$ similarity to veins and a dissimilarity to the protolith [*Techer et al.*, n.d.; *Clauer et al.*, 2017; *Mazurek and De Haller*, 2017]. Our finding of non-porous, µm-thin shear zone walls relates to several studies reporting fault-parallel fluid-flow, wherein





permeabilities perpendicular to the fault are several orders of magnitude lower than parallel to it [e.g. *Arch and Maltman*, 1990; *Caine et al.*, 1996; *Crawford et al.*, 2008; *Warr et al.*, 2014].

### 4.2.2 Stage 2 – development of type 1 gouge from scaly clay

*Development of P-foliation*

Numerous studies propose a passive reorientation of grains between movement-accommodating principle shear zones [*Morgenstern and Tchalenko*, 1967; *Cladouhos*, 1999a, 1999b; *Haines et al.*, 2013], i.e. grains rearrange towards a P-orientation due to offset along C- and Y-shears [cf. *Lin*, 2001; *Bigi*, 2006].

*Calcite dissolution*

The reorientation process is proposed to comminute (fossil) calcite between the rearranging grains and this promotes the calcite

dissolution by fluids, perhaps related to a temporary permeability increase by local dilation in restraining bands (similar to Figure 14e and Figure 15e) until a static, high intensity foliation is reached and the fault is re-sealed [cf. *Takizawa and Ogawa*, 1999; *Holland et al.*, 2006; *Bock et al.*, 2010]. Moreover, calcite dissolution is enhanced by the onset of grain size reduction [*Rutter and Elliott*, 1976; *Gratier et al.*, 2014].

*Generation of nm-sized clay grains*

Matrix material, i.e. nm-sized clay grains (Figure 15b), is more abundant in gouge than in undeformed OPA. It may be generated by cataclastic delamination processes and/or by neoformation. A model for authigenic gouge evolution by fluid flow along pre-existing shears was also proposed elsewhere [e.g. *Solum*, 2003; *Schleicher et al.*, 2010; *Janssen et al.*, 2013; *Haines et al.*, 2014; *Warr et al.*, 2014; *Buatier et al.*, 2015]. In experiments, cataclastic generated gouges yield power-law GSDs with exponents (D) increasing with strain until a static value of D ~ 2 is reached. Henceforward, abrasion and wear dominate over

grain splitting [e.g *Sammis and King*, 2007; *Mair and Abe*, 2011]. For OPA, undeformed material shows D = 1.97 already in existence, in parity with the gouge (D = 1.98). Accordingly, gouge might not develop a larger D-value with strain but only an increase in smallest grains by abrasion. The GSD of OPA gouge is inferred to increase in D for grains below the resolution limit, i.e. grains < 10,000 nm² have a large contribution to the total GSD. Contrary, for synthetic quartz gouge, Keulen et al. [2008] report a decrease in D for grains below 1.1 µm radius [so-called "grinding limit", cf. *Kendall*, 1978]. This discrepancy,

combined with geochemical results [*Clauer et al.*, 2017, see above] suggests that neoformation contributes significantly to the generation of nm-sized clay grains, even though abrasion and delamination of larger grains occurs too (cf. Figure 16).

*Origin of amorphous SiO₂*

The origin of amorphous $SiO_2$ grains that we found in the gouge (Figure 15) remains enigmatic. Either it formed from Si-rich fluids or it is of detrital origin, smeared from a source layer. However, mineralogical analysis have not found any amorphous

material in OPA [*Lerouge et al.*, 2014, see above]. Potentially, the grains were rounded during the gouge evolution: Kameda et al. [2017] consider that hydrous opal-CT is highly deformable via pressure solution creep at rather low driving stresses in the order of kilopascal. This consideration might hold true for the rheology of the boudinaged $SiO_2$ grains in Figure 13.



Stage 2 results in gouge type 1 fabric that is illustrated by Figure 8, showing the P-orientation of mineral foliations and (not yet reworked?) host rock clasts. Note, that the clasts are separated along P-oriented dark gouge: an indicator for progressive rework of host rock into gouge material.

### 4.2.3 Stage 3 –development of type 2 gouge

Similar to scaly clay aggregates, an S/C fabric is visible in developed gouge type 1, too (Figure 4). This fabric suggests a continuing application of the processes of stage 2 to generate the more mature type 2 gouge, with the dissolution of all remaining host rock clasts and further dissolution of calcite. Once developed, the P-foliation angle to shear direction is not a result of simple shear strain (i.e. offset), but of the amount of pure shear [*Cladouhos*, 1999a, 1999b]: volume loss by dissolution of calcite potentially resulted in a decrease of shear zone width, lowering the P-foliation angle. Thus, the differing dissolution
of calcite likely causes the differing P-foliation of both gouge types (Figure 4b). From geochemical analysis, Clauer et al. [2017] propose gouge exclusive sulfate and/or phosphate-rich fluids that enhance the localized calcite dissolution in gouge. We propose that the gouge development is more effective in the early stage 2, followed by a change in micromechanical behavior, where frictional granular flow of the existing gouge material and sliding along µm-thin R- and Y-shears dominate over further rework of the host rock [cf. *Guo and Morgan*, 2007].

### 4.3 Physical rock properties of gouge

The strong clay grain alignment in µm-thin shears suggests a low residual friction coefficient along (001) lattice planes once faulting is initiated [cf. *Collettini et al.*, 2009; *Mizoguchi et al.*, 2009]. Ultra-fine clay particles, the strict P-foliation, the strong calcite reduction, the strained but not fractured organic matter and the well-rounded amorphous $SiO_2$ grains in gouge emphasize a low-rate dependent, viscous deformation with distributed brittle shear along R- and Y-shears. By BIB-SEM, we find the
porosity of gouge to be lower than porosity in undeformed OPA (< 18 %). However, a large population of smallest pores beyond the resolution limit of this method (< 10 nm) is considerable. The µm-thin shears are found non-porous, even for pores < 10 nm in the TEM, potentially enabling a trap of high-fluid pressures inside gouge [cf. *Zhang et al.*, 2001; *Cuss et al.*, 2011]. Such local high fluid pressures promote further strain localization, which, likely, occurred progressively, due to the interplay of pore pressure generation, tectonic stress built-up and (slow) pore pressure dissipation. Contrary to rather brittle behavior of
OPA in most laboratory experiments, the in-situ gouge fabric suggests a long-term creep behavior and distributed shear along µm-thin shear zones [cf. *Giger et al.*, 2008], where fluid flow is drastically inhibited perpendicular to shear, but temporarily enhanced parallel to shear inside the gouge. In direct shear experiments, Bakker et al. [2017] showed for synthetic OPA gouges that a clay-enriched gouge has a lower friction coefficient than gouge with an unaltered OPA composition. Thus, as the gouge from the Main Fault shows a high content of clay matrix compared to the undeformed OPA, we expect the natural gouge to be
mechanically weak, i.e. the gouge requires a lower shear stress to induce strain than the undeformed OPA.





## 5 Conclusion

We described the microstructure of natural, upper-crustal fault gouge from the low-offset Main Fault in the Mont Terri Research Laboratory (CH), mainly using optical and electron microscopy techniques. We inferred a complex interplay of deformation mechanisms and thus provide insight to the low-rate in-situ deformation behavior in OPA. Our findings can be summarized as follows:

1. Gouge sharply stands out as a dark (calcite-poor), thin (< 2 cm), continuous band at the upper Main Fault boundary and as isolated patches within neighboring scaly clay aggregates. It comprises two fabric types, differing in P-foliation, that both show internal µm-thin, non-porous shear zones in R and Y orientation. Gouge has a higher amount of clay matrix than undeformed OPA. By TEM, the gouge matrix consists of nm-sized clays with a random orientation, while larger grains show a strong fabric intensity in P-foliation. We found amorphous, well-rounded $SiO_2$ in the gouge. The SEM-visible porosity of gouge is strongly reduced compared to undeformed OPA, however an increased porosity for gouge by pores smaller than the SEM resolution limit [cf. *Orellana et al.*, 2016] is possible.

2. We found indicators for all major deformation mechanisms, including neoformation of clays, pressure-solution of calcite, cataclasis and abrasion as well as frictional granular flow. The interplay of these mechanisms suggest a low-rate, progressive deformation of gouge. Our microstructural interpretations of deformation mechanisms are in concert with recent geochemical solute-transport studies [*Clauer et al.*, 2017; *Mazurek and De Haller*, 2017].

3. We present two very different hypotheses for the evolution of the gouge: (1) fluid-mediated reworking of undeformed OPA and/or (2) clay smear from a (yet to be identified) calcite-poor source layer. We favor hypothesis (1), as the S/C fabric in gouge type 1 suggests a progressive rework of the protolith. This hypothesis would be supported by the occurrence of transition stage microstructures, which, however, are not clearly recognizable. A combination of both hypotheses is considerable, too.

4. We consider that in-situ, the clay-enriched gouge is weaker than undeformed OPA and exhibits a viscous rheology with distributed shear along µm-thin shear zones.

Future work on OPA gouge should include rotary shear tests of undeformed OPA to record the fabric development and rheology at large strains. The result could be compared with our findings to infer the impact of processes that are not fully reproducible in the laboratory, such as fluid interactions and low strain rate.

### Acknowledgements

For the well-accomplished XRD analysis, we thank Uwe Wollenberg and Pieter Bertier, who also helped discussing the XRD methods and results. We thank Arne Grobe for providing the VR and TOC measurements. Norbert Clauer and Isabelle Techer helped by providing valuable geochemical data of the study area and by discussing fluid-rock interactions. Jan von Harten and Wiebe Förster are thanked for their excellent help in particle segmentation. Swisstopo and the Mont Terri Consortium are thanked for continued interest and financial support.



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



## Figures

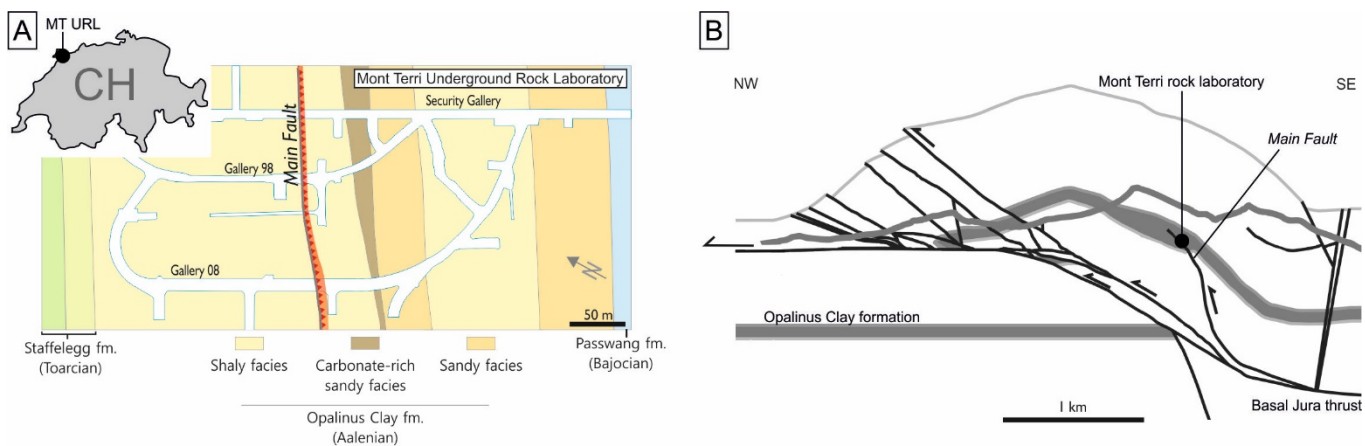

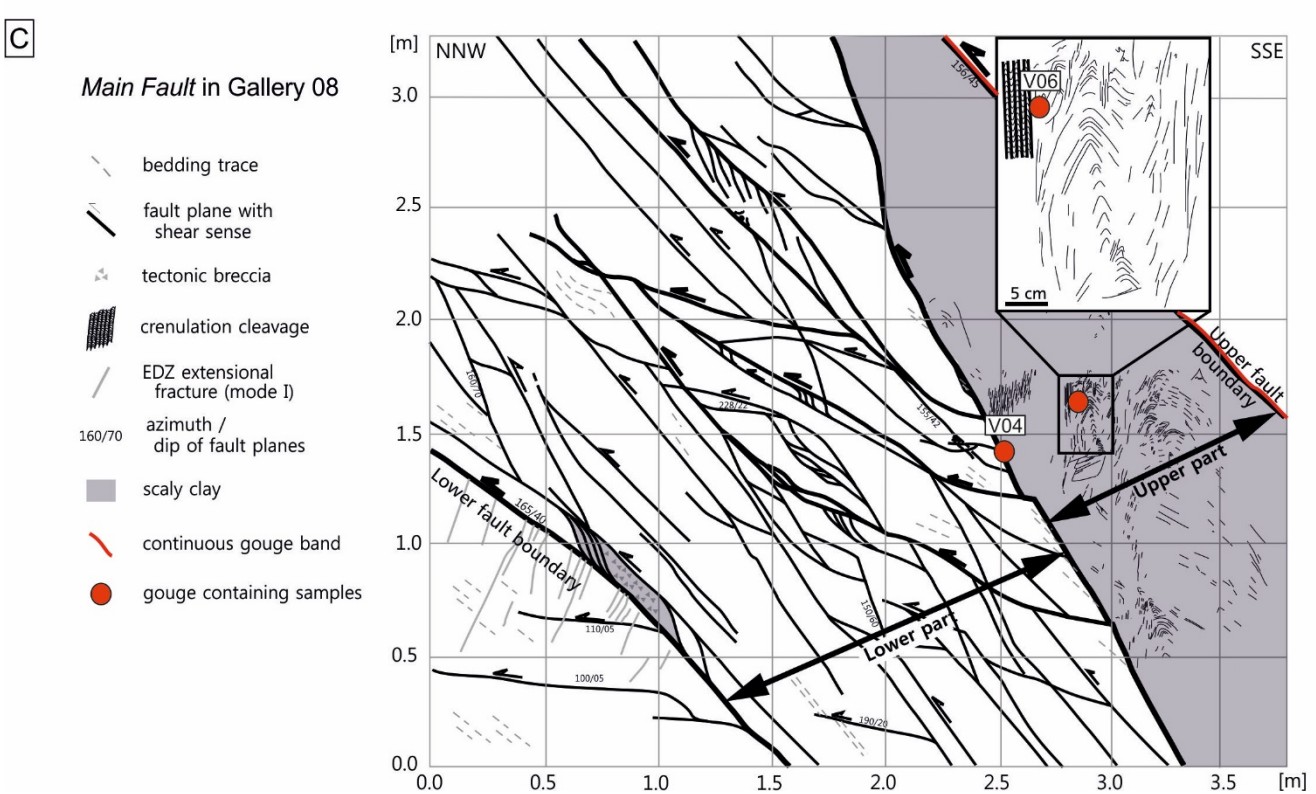

Figure 1: (a) location map of the Main Fault in the Mont Terri research laboratory, (b) cross-section modified from Freivogel et al. [2003] and (c) outcrop sketch of the Main Fault in gallery 08 modified from Nussbaum et al. [2011].

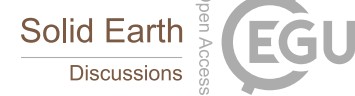


**Figure 2: Outcrop picture of the continuous gouge band at the upper fault zone boundary in gallery 98 (view to ENE).**



**Figure 3: Illustration of (i) the sharp gouge (dark) to wall rock (bright scaly clay) contact, (ii) the distinct boundary between type 1 and 2 gouge, (iii) the gouge internal P-foliation and (iv) gouge-internal clasts, as well as (v) surrounding calcite veins. (a) shaded light photograph of water-immersed hand-specimen V15 with (b) sketch and (c) SEM-EDX Ca distribution map of the sample. Note that gouge is drastically reduced in Ca compared to scaly wall rock. (d) and (e) are insets of (a). See text for details.**





**Figure 4: P-foliation patterns, grain and wall rock clasts as well as µm-thin shear zones in gouge. (a) water-immersed hand-specimen V17 photographed in shaded light. (b) inset of (a), (c) sketch of (a). Note the S-C resembling sigmoidal P-foliation and the strict boundary separating type 1 and 2 gouge. See text for details.**



**Figure 5: (a) top-view on broken shear zone surfaces (slickensided surface) in gouge (sample BPS12-4-S1). (a) reflected light photograph, (b, c) SEM (SE) micrographs displaying nm-sized clay particles on a smoothly polished slickensided surface. (d) SEM-EDX Ca distribution map of (c). Contrary to shear zones in undeformed OPA, there is no Ca present, not even at slickenside risers.**




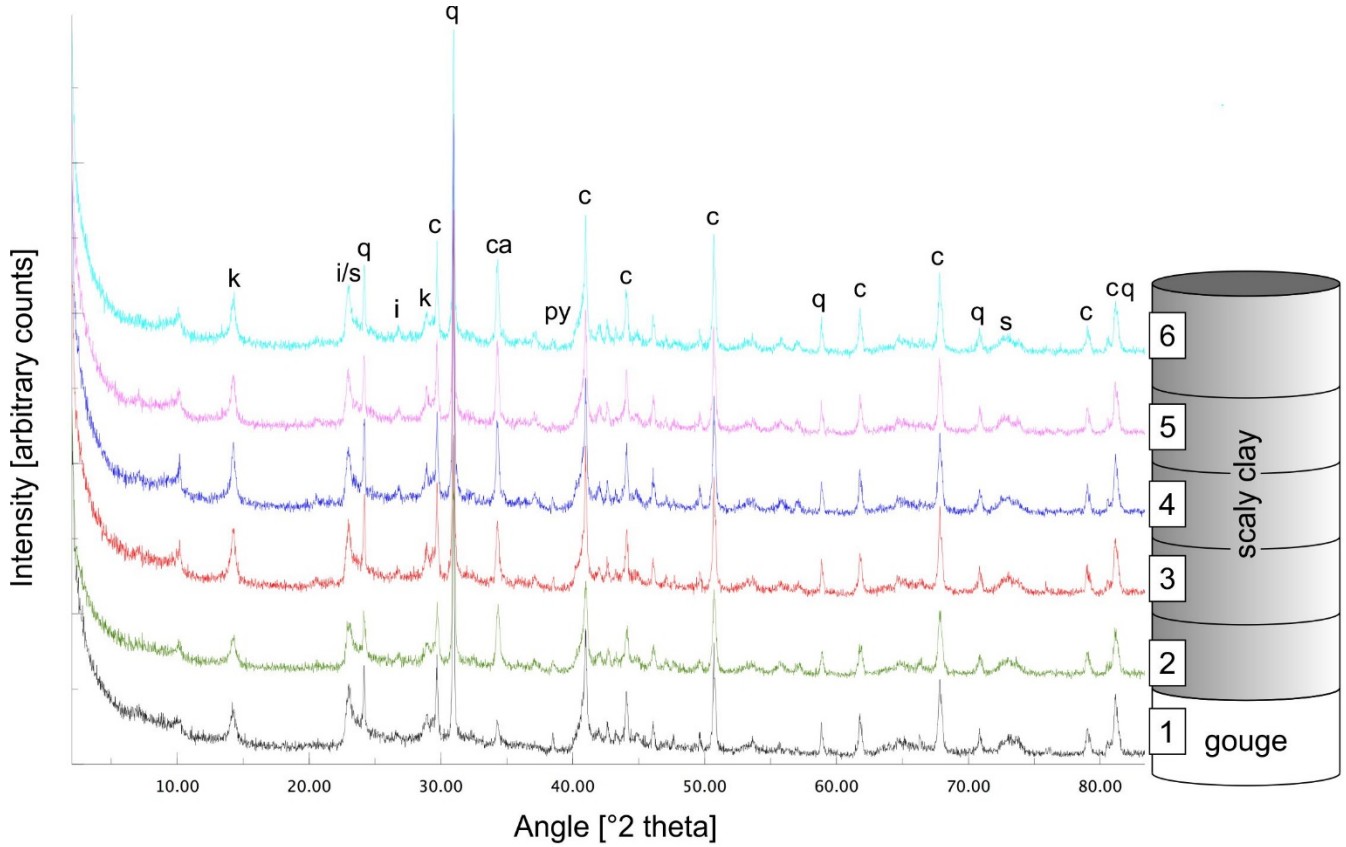

**Figure 6: Stack of six XRD pattern from sample BPS12-4. Pattern 1 belongs to a gouge layer, pattern 2 is 1 cm from gouge, pattern 3 2 cm and so forth (sketched). Note that there is no obvious change in the pattern except a reduced calcite peak in gouge. Measurements were done on a Huber MC9300 diffractometer using CoKa-radiation produced at 40 kV and 40 mA. k = kaolinite,**
5    **I = illite, s = smectite, q = quartz, ca = calcite, py = pyrite, c = corundum (20 wt.% added as internal standard).**



**Figure 7: Series of shear zones within gouge and surrounding wall-rock (scaly clay) as well as host rock clasts. (a, c and d) reflected light micrographs, (b) sketch of (a), (d) inset of (c). Dashed lines and arrows mark the wavy net of dark shear bands, which are wider than μm-thin shear zones (cf. Figure 10). (a, b) sample A3-1 polished with 800 grid grinding paper, (c, d) sample V15-T2 polished by large-area BIB. The images are contrast enhanced to make the black bands stand out. Best viewed in a high-quality color print or in magnified pdf. See text for details.**



**Figure 8: Host rock clasts in gouge type 1. (a) reflected light micrograph and (b) sketches of sample A9c-1 polished by large-area BIB. (c) shaded light photograph of water-immersed sample V04 showing foliation. Note the sigmoidal S/C like pattern of the clasts, which suggests a progressive rework of the host rock into gouge. See text for details.**



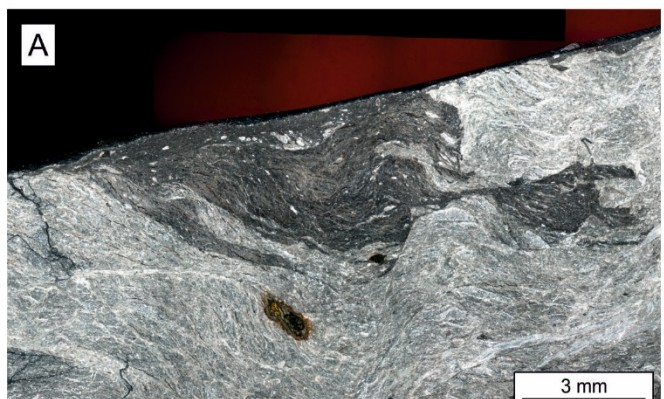

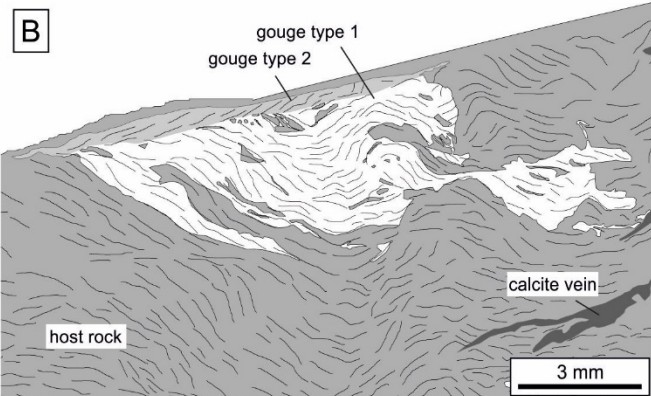

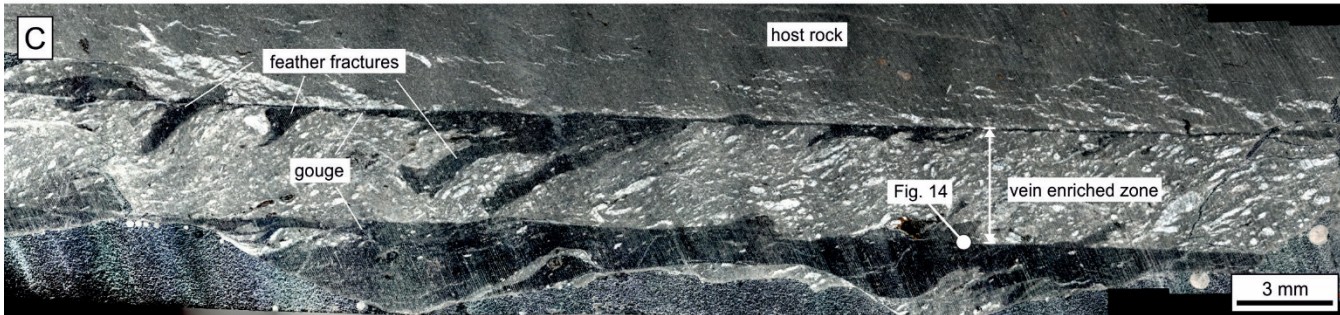

**Figure 9: Mixing-like textures of gouge into scaly wall rock. (a) reflected light micrograph and (b) sketch of sample V06. (c) reflected light micrograph of sample A1. Note that the feather fractures are synthetic; the shape of wall rock clasts in (a) follow the gouge foliation. A band of gouge type 2 is persevered at the upper border of the sample in (a). See text for details.**





**Figure 10: Comparison of undeformed OPA (right, sample V08) to gouge fabric (left, sample V15-4) with grains in P-foliation and with a 30 µm wide shear band of differently oriented grains. All micrographs are BIB-SEM (BSE) recorded, (b) and (e) are insets of (a) and (d), respectively. (c) and (f) manually segmented grains used in statistical analysis (cf. Figure 11). Zoom-in for full detail.**



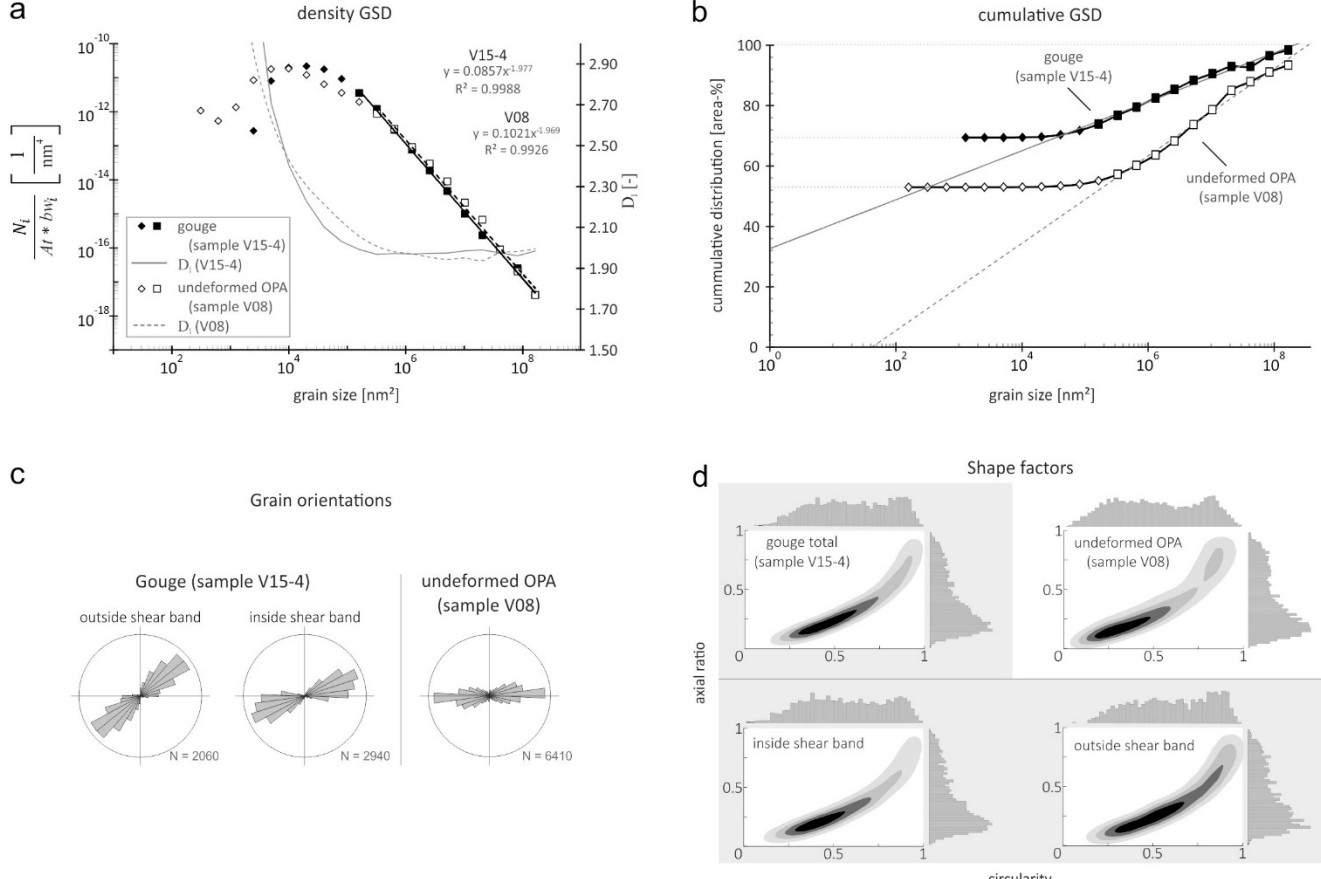

**Figure 11: Statistics on grain size, shape and orientation of samples V15-4 (gouge) and V08 (undeformed OPA), cf. Figure 10. (a) density grain size distribution (GSD): grain size vs. number of grains ($N_i$) noted with respect to bin width ($bw_i$) and normalized for the examined area ($At$); $D_l$ = local slope, indicating a deviation from the power-law trend for smaller grain size bins due to resolution limitations (truncation). Boxes: values used in power-law trend calculation, diamonds: values excluded from power-law calculation. Note the parity in density GSD for undeformed OPA and gouge. (b) cumulative GSD: grain size vs. cumulative area-%. Note that the distribution does not sum-up to 100 area-% due excluding (i) pores and (ii) grains that do not fall completely within the examined area. The curves start depending on the matrix content of the samples, i.e. on the area-% that is not visually segmentable. Note that a strict extrapolation for the cumulative gouge GSD results in unrealistic small grains (less than one lattice plane distance). Hence, a larger contribution of grains below the BIB-SEM resolution limit is inferred. (c) grain orientations weighted by the grains' axial ratio. (d) axial ratio vs. circularity, with the mean center of gouge deviating from undeformed OPA, indicating an increased smoothness of the gouge grains.**



**Figure 12: Comparison of undeformed OPA (right, sample V08) to gouge porosity (left, sample V15-4). All micrographs are BIB-SEM recorded (SE-detector). (c) and (f) same location as (b) and (e) recorded with BSE detector. Note the decreased visible porosity in gouge. Larger cracks in gouge are interpreted as artifacts. Bright grain left in (e) and (f) is part of a calcite fossil. Zoom-in for detail.**



**Figure 13: Details of gouge microstructure (sample V15-4, cf. Figure 10a, b) with strict foliation discordance (sharp boundary of shear band to P-foliation domain), trans-granular fractures, boudinaged grains, strained organic material (OM), µm-thin shear zones (b) and ultra-fine grained matrix. All micrographs recorded with BIB-SEM (BSE).**



**Figure 14:** Microstructures of a gouge-internal shear band (sample A1-1) with µm-thin Y-shear boundaries (d), strained organic matter (dark grains), and pores in pressure shadow regions of larger grains (e). (a) reflected light micrograph and (b) sketch of foliations. (c, d and e) BIB-SEM (BSE) micrographs. The shear band is located at the border of gouge to vein-enriched scaly wall rock. See Figure 9c for location.

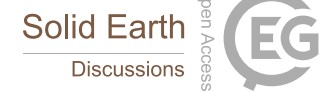

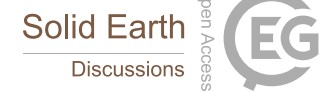

**Figure 15: Microfabric of gouge (sample V15-FIBgouge1) visualized with TEM methods. (a) HAADF micrograph containing a gouge-internal μm-thin shear zone (e), P-foliation (d), pores in pressure shadow regions, e.g. (e) and cataclastic kink in (d), rounded, amorphous SiO₂ grains (also Figure 16), a dragged mica and un-resolvable small, ultra-fine grained gouge matrix (b). (c) EDX and SAED from region shown in (b), indicating that the matrix is polycrystalline clay. See text for details.**





**Figure 16: Microstructural details of gouge shown in Figure 15 (sample V15-FIBgouge1). (a) TEM (HAADF) micrograph and SAED pattern of amorphous SiO₂. (b), (c) and (d) HR-TEM (BF) insets of (a) showing delamination and breakage of a mica(?) grain. See text for details.**



**Figure 17: Generic sketches of gouge type 1 (top) and type 2 (bottom), summarizing all found microstructural elements [after**
***Laurich*, 2015]. a transpressional folding of foliations (occasional). b sigmoidal foliation (resembling S/C fabric), fabric intensity**
**higher than in other areas of scaly clay. c intense uniform P-foliation inside gouge domains (always). d clast (pyrite, quarz, feldspar,**
**calcite) parallel to sub-parallel to P-foliation. e spalled fragments of clast (often with calcite clasts). f/i/j localized shear bands, always**
**synthetic: (f) in R-orientation, rare in gouge with low angle P-foliation, (j) in Y-orientation, (i) foliation parallel to gouge domain**
**boundary (always if boundary is shear sense parallel). g gouge lens inside host rock inclusion (occasional). h strained host rock**
**inclusion (often). k strained and fragmented vein in R-orientation (occasional). l scaly clay fabric with anastomosing foliation (no**
**S/C foliation). m type 2 gouge domain with P-foliation sub-parallel to shear sense, no host rock inclusions, more clasts than in (n). n**
**type 1 gouge domain with P-foliation oblique to shear sense, does contain host rock inclusions. o S-C fabric (common in type 1 gouge,**
**rarely in type 2). p vein parallel to gouge boundary (occasional). q scaly clay with foliation in P-orientation.**



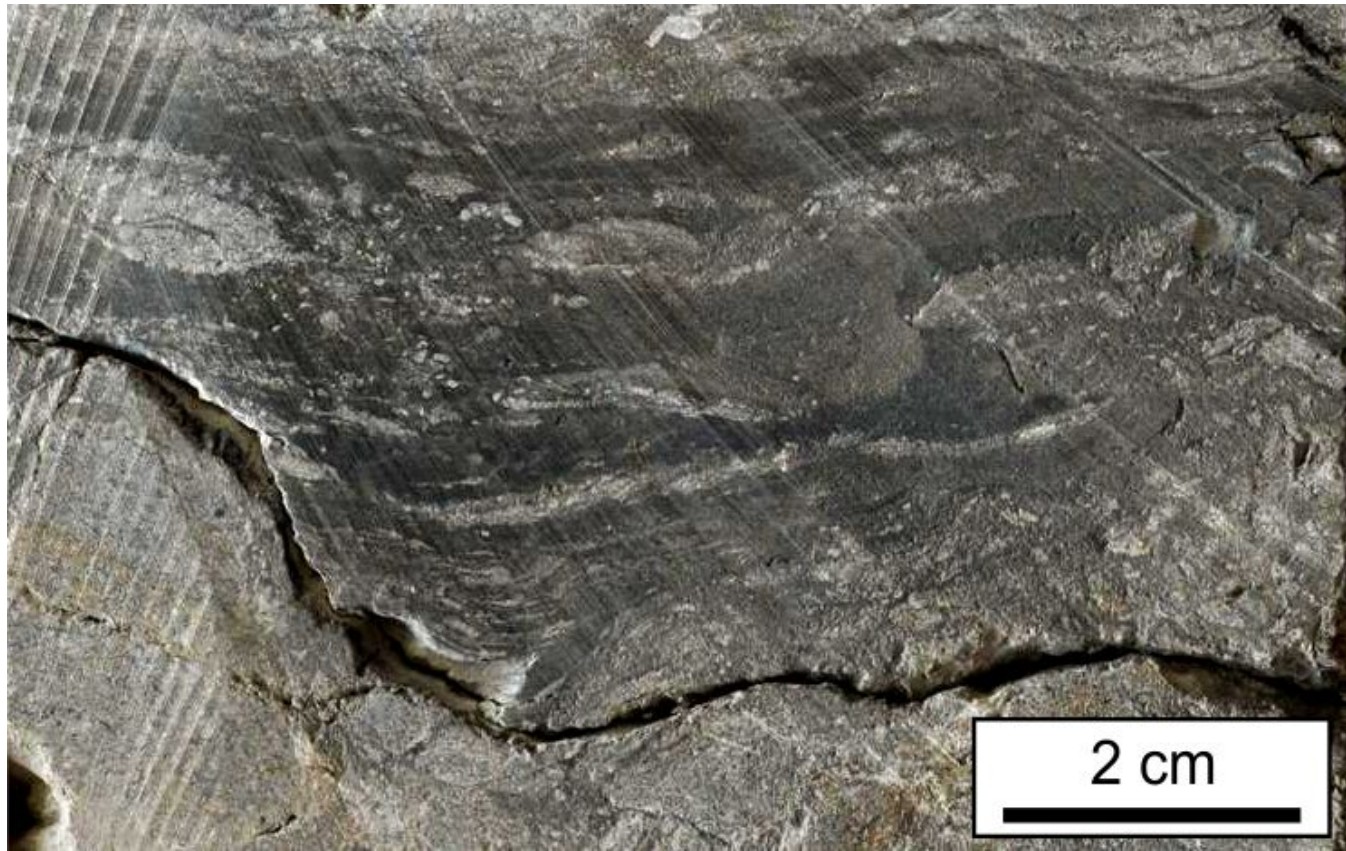

**Figure 18: Photograph of dark, macroscopically gouge-similar, bedding parallel bands from drill core BDB-1. This is to indicate that, contrary to the hypothesis of authigenic gouge generation, gouge could be allogenic, being smeared-in from such dark bands. Next to the Main Fault, however, those bands are not reported. See text for details.**