# Peer review of "Deformation mechanisms and evolution of the microstructure of gouge in the Main Fault in Opalinus Clay in the Mont Terri rock laboratory (CH)"

_Solid Earth, 2017_

## Short Comment (SC1) · 26 Jun 2017

The paper is a very good work concerning structural and microstructural evolution of foliated clay-bearing gouge along a thrus fault. In particular, the paper investigate in depth the deformation mechanisms acting both at microscale and at nanoscale.

Watching the Fig. 9, could be possible that the intruding black structures with irregular and convolute margins are fluid-like structures? they look very similar to those observed by Brodsky et al., (2009), Demurtas et al., (2016) and interpreted as generated

by high-fluid pressure (Fondriest et al., 2012). Maybe such structures could be the result of fluid overpressure within such low-porosity gouge. The occureence of over-pressured fluids was also suggested by the author at pag. 11 (line 24) and at page 13 (line 22).

Moreover, the nano-size clay spherules observed along sharp slip surface (Fig. 5) could also contribute to make the gouge far weaker than the host rock, as recently suggested by Chen et al., (2017, and reference therein, in Fault Zone Dynamic Processes: Evolution of Fault Properties During Seismic Rupture, Geophysical Monograph 227) for nano-particles occurrence within gouges. Maybe this point could be strengthened reinforcing the point stated by the author of a very weak gouge.

Thank for this excellent contribution.

Best regards, Luca Smeraglia
* * *

---

## Referee Comment (RC1) · Anonymous Referee #1 · 20 Jul 2017

**Review**: Deformation mechanisms and evolution of the microstructure of gouge in the Main Fault in Opalinus Clay in the Mont Terri rock laboratory (CH) by Laurich et al.

General comments

This is a well-documented, profound and useful paper. The figures are all well done. Many images are presented, which helps the reader to understand such nano-scale microstructures. The quantitative microstructural data are well performed and include high-end techniques. The bulk rock data are standard techniques and may have not an high impact on the discussion and conclusion of the paper. Some of these may also exclude.

I have no fundamental changes to the science of the manuscript.

Some detail comments:

Page 4,   Line 16: You may delete "beautiful" (not scientific in this context).

Page 5,   Line 25: What is DIA?

What is the control on segmentation? Due to only BSE contrast between different mineral-types, the GSD of the bulk aggregate may be biased by the type of used contrast? You may comment on this in the "Method"-section. You may include some additional reference for the relationship between the GSD and the particles classified as matrix (e.g., Heilbronner and Keulen 2006, Keulen et al. 2007). The relation between matrix and clasts is a known fabric parameter (see discussion in Hadizadeh and Johnson 2003)

Page 7,   Line 1-5: Is this not better presented in the chapter "Methods"?

Line 11ff: The comparison of different methods (Mercury porosimetry, He-pycnometry and Image Analysis) are often not direct comparable (e.g., connected versus isolated pores, 2D versus 3D, etc). I propose to omit this sentence. Alternatively, you may write more in detail for this comparison.

Page 8,   Line 22: Do you have other $SiO_2$ modifications as quartz? You discuss this later, but should be mentioned before.

Page 10,   Line 1: You may add: "… suggesting intracrystalline plasticity *or fracturing parallel (001)*."

You may combine chapter 3.3.1 and 3.3.4. These two processes are somehow connected. You produce smaller grain sizes by cataclasis, which allow in the following frictional flow. This would be easier to read.

Page 11,   Line 28-29: The argument using the Sr-isotope data is not understandable. I would always expect different Sr-data in veins, water and protholith caused by fractionation between different minerals (i.e. calcite versus clay) and the main difference between protholith and vein is the mineralogy. You may explain better, what is the argument (or omit this argument).

Page 12,   Line 28ff: Many studies indicate that amorphous $SiO_2$ is not stable over geological times above a certain temperature. Many arguments have been found, that amorphous $SiO_2$ are precipitated and reorganized into quartz in geological time scale. Your findings, may discussed in this context: (1) is the amorphous $SiO_2$ geological developed?; (2) what are T-t conditions to stay amorphous?

A personal comment:
The paper has many abbreviations (e.g., BIB_SEM, SAED, OPA, XRD, TOC, VR, DIA, GSD, etc.), which is sometimes difficult to read. This would be even more difficult, for a reader, which is not from the same scientific community.

Possible Additional references:
Hadizadeh and Johnson (2003), Estimating local strain due to comminution in experimental cataclastic textures. J. Struct. Geol. 25, 1973–1979.
Heilbronner, R., Keulen, N., (2006). Grain size and grain shape analysis of fault rocks. Tectonophysics 427, 199–216.
Keulen, N., R. Heilbronner, H. Stunitz, A.-M. Boullier, and H. Ito (2007), Grain size distributions of fault rocks: A comparison between experimentally and naturally deformed granitoids, J. Struct. Geol., 29(8), 1282–1300.

---

## Referee Comment (RC2) · J. Hadizadeh (Referee) · 25 Aug 2017

Laurich et al. have presented a reasonably well-written manuscript with a wealth of microstructural information on clay-rich gouges from a relatively young fault zone. In this sense it is an interesting contribution to our understanding of clay-rich gouges, which are often associated with more mature fault zones. The authors use a variety of analytical techniques and microstructural analysis all of which are adequately described. Based on this review and considering comments by two other reviewers, I believe the authors should make some relatively minor changes before the manuscript is ready for

publication in Solid Earth.

1. The proposed hypotheses: The authors arguments regarding the "smear" hypothesis are vague at best. For example, why is it necessary for the possible source of the smear to be calcite-free (line 6 in 4.1)? Do the authors suggest the "smear" hypothesis as an alternative explanation only for the existence of the dark, calcite-free gouge type? What would be some relevant and expected characteristics of such "smeared-in" gouge? This hypothesis may even be considered untestable if a source for the smear cannot be identified, or is so highly speculative. The "authigenic generation" and/or "reworking of OPA" hypothesis on the other hand, is supported by evidence of progressive deformation (sharp difference in calcite content between the two gouge types; continuously traceable reduction in Riedel shear angles with rescept to shear zone borders, going from the calcite-rich gouge to the dark gouge; low clast-matrix ratios indicating higher strains in the dark gouge). As the authors have noted, different microstructural domains in the gouge indicate a number of deformation mechanism transitions that lead to relatively low-friction localization zones. This alone, questions the possibility of a "smearing" process since under steady P-T conditions smearing implies low friction of the smeared material. I agree with the authors that pure shear has been a factor in late development of the gouge layer geometries both due to change in deformation mechanism toward a less dilatant behavior as well as volume change. Such processes are more likely to be in-situ (or "authigenic" as the authors put it) as opposed to resulting from microstructural evolution of a "smeared" gouge. The evidence of amorphous SiO2, which is clearly presented in Figs. 15 and 16 of the manuscript, is consistent with late stage hydrothermal SiO2 in the gouge in agreement with comments by the anonymous reviewer regarding the stability of amorphous silica with T and age of the gouge. I suggest that the "smearing" origin of the studied gouge, as a distinct hypothesis, be removed. Given the authors current information, one may only include the possibility of spatially limited smearing event(s) within framework of the reworking hypothesis.

2. To strengthen the reworking hypothesis the authors need to strengthen the arguments that support OPA to scaly-clay transitional stage during progressive deformation. For this, they may need to carefully reexamine Figs 2, 3, 4 for microstructural relationships between scaly wall rock (OPA) and the rest of the gouge types and chemical map of calcite related to these boundaries. The section 4.2.1 describing the transition from OPA to scaly clay appears to me inadequate in view of its significance as mentioned above.

3. In agreement with comments by anonymous reviewer about comparison of porosity measurements obtained via different methods, the use of mercury porosimetry in particular (cited estimates by the authors) is controversial because of uncertainties in non-fracture porosity values (usually overestimations) caused by elastic deformation of pores of different size. Estimates via careful image analyses is more reliable than this method. However, the authors have left readers to eyeball an estimate of the gouge porosity by looking at very few images in the manuscript (lines 5-8, P7), which could vary depending on the reader's experience with microstructural porosity. From Fig. 12e it is possible to suggest that higher porosity of OPA is due to primary porosity of mineral fragments (e.g. calcite and quartz) in the clay matrix.

4. Other comments: –I strongly agree with the anonymous reviewer about combining sections 3.3.4 (frictional granular flow) and 3.3.1(cataclasis and abrasion).

–In Fig. 3B, traces of internal shears seem to have a light green color. If this is not a compositional color coding, it is better represented in thin black lines.

–Fig. 8 lacks description for part D.

–Fig. 17e- where you have "(often with calcite clasts)", do you mean: common in calcite clasts?

---

## Author Comment (AC1) · 23 Oct 2017

**Answer to the reviewers**

In total, we received 3 reviews, of which one is an interactive comment by Luca Smeraglia and two are regular referee comments by Jafar Hadizadeh and an anonymous reviewer. The referees rated our contribution as "to be published with minor changes". Their encouragement and their precise, expert comments led to an improvement of the manuscript. Thank you very much!

Below, we present our manuscript changes with respect to the reviewers' main comments. Page and line references relate to the resubmitted manuscript, if not explicitly stated otherwise. This holds true for the changed Figure numbering.

Moreover, all changes are highlighted in the resubmitted manuscript.

**Interactive comment by Luca Smeraglia (University of Rome)**

The reviewer has two main comments on the manuscript:

1. The 'injection' appearance of gouge in Figure 5 (formerly Figure 9) might be due to fluid overpressure in the gouge.
   We agree, that the pattern in Figure 5 resemble an 'injection' appearance. This is already stated (p.9, l.7). We thank the reviewer for the hint to further literature on this matter and added useful references he recommended (p.9, l. 7ff). Moreover, we highlight the connection to fluid overpressure and the possibility that this could promote such patterns. Furthermore, we complemented the generic microstructure sketch of gouge (Figure 17) by the injection pattern.
   New / modified text:
   *"The preserved high fabric intensity (Figure 4), the well-developed S/C fabric pattern (Figure 6) and the occasional 'mixing' and 'injection' appearance of gouge (Figure 5, cf. cataclasites in Chester et al. [1998], Brodsky et al.[2009], Demurtas et al. [2016]) suggest a continuous granular flow, where individual particles are aligned in response to movement along shear bands and µm-thin shear zones."*

2. The nano-size clay spherules in Figure 14 (formerly Figure 15) might have a contribution in weakening the gouge compared to the host rock.
   We do fully agree on this point, especially in connection with water that could be adhesive to the nano-sized particles. This hypothesis is stated already (p.9, l.13). However, we consider a weakening by "particle rolling", as in high velocity tests by Chen et al. (2016) and Han et al. (2011), as unlikely. We modified the paragraph by stating that the gouge matrix is weaker due to a lower viscosity, but that strain is still accommodated by frictional offset along face-to-face aligned clay particles in µm-thin shear zones (p.9, l.14).
   New / modified text:
   *"We therefore propose that the ultra-fine matrix was significantly weaker than the rest of the gouge, as its small, likely less elongated particles are easy to reorient, possibly aided by water bound to the nm-grains. This setting lead to a complete loss of preferred orientation of nanoparticles in parts of the gouge (Figure 14c). Yet, '(nano)particle rolling', as found in high-velocity tests of synthetic gouge [Han et al., 2011; Chen et al., 2017], might not had a large impact on gouge weakening in the low-velocity Main Fault. Instead, the matrix likely responded with a lower viscosity to frictional offset along face-to-face oriented particles in µm-thin shears [cf. Jessell et al., 2009]."*

**Referee comment by an anonymous reviewer**

1. General comment:
   "The bulk rock data are standard techniques and may have not an high impact on the discussion and conclusion of the paper. Some of these may also exclude."
   We like to briefly state in the paper that we have performed such techniques and that they underline a similarity of gouge material to undeformed OPA. This result does not help to rule out the "clay-smear" or the "reworking" hypothesis for the gouge material and thus the bulk rock data indeed do not have a large impact on the discussion. However, deleting these results, the reader might wonder if such data could bear valuable information (section 3.2).

2. Page 4, Line 27: You may delete "beautiful" (not scientific in this context).
   We agree and killed this statement ☹. (p.4, l.27)

3. Page 6, Line 4 (formerly p.5, l.25): What is DIA? What is the control on segmentation? Due to only BSE contrast between different mineral-types, the GSD of the bulk aggregate may be biased by the type of used contrast? You may comment on this in the "Method"-section. You may include some additional reference for the relationship between the GSD and the particles classified as matrix (e.g., Heilbronner and Keulen 2006, Keulen et al. 2007). The relation between matrix and clasts is a known fabric parameter (see discussion in Hadizadeh and Johnson 2003)
   We meant "Digital Image Analysis", but deleted this wording and hope that it is easier to read now. (formerly p.5, l.25, resp. p.6, l.4)
   Yes, the manual segmentation is based on the BSE contrast. When recording both micrographs, we strived for similar contrast with highest detail. This was mentioned (p.6, l.19) and is now also mentioned in the methods section (p.3, l.21).
   We noted the differences in matrix content for both samples (p.8, l.8) and added now the clast/matrix ratio for each sample (p.8, l.8). However, we think that for our case the ratio is not suitable to evaluate the deformation intensity, at least not as done in the suggested literature, which are on cataclastic produced gouges. This is because of three arguments: (1) the protolith is already fine-grained with a D value for segmented grains in parity with those determined for gouge (p.6, l.16 and p.13, l.4), (2) nano-particles might be neoformed by fluid precipitation and (3) other deformation mechanisms (offset along μm-thin shear zones, viscous behavior) are likely dominating over grain fracturing. This argumentation, in reasoning to the suggested literature, is stated in (p.13, l.5ff) with a reference to Keulen et al. 2007:
   *"…gouge might not develop a larger D-value with strain but only an increase in smallest grains by abrasion. The GSD of OPA gouge is inferred to increase in D for grains below the resolution limit, i.e. grains < 10,000 nm² have a large contribution to the total GSD. Contrary, for synthetic quartz gouge, Keulen et al. [2008] report a decrease in D for grains below 1.1 μm radius [so-called 'grinding limit', cf. Kendall, 1978]. This discrepancy, combined with geochemical results [Clauer et al., 2017] suggests that neoformation contributes significantly to the generation of nm-sized clay grains, even though abrasion and delamination of larger grains occurs too (cf. Figure 16)."*

4. Formerly Page 7, Line 1-5: Is this not better presented in the chapter "Methods"?
   We agree and moved these lines to the methods section (now p.3, l.25ff).

5. Page 7, Line 18ff: The comparison of different methods (Mercury porosimetry, He-pycnometry and Image Analysis) are often not direct comparable (e.g., connected versus isolated pores, 2D

versus 3D, etc). I propose to omit this sentence. Alternatively, you may write more in detail for this comparison.

We agree that the methods can yield differing results. And because of that we think it is of value to cite the work by Orellana et al. (2016), which presents such a differing result. Plus, the work by Orellana et al. (2016) is the only alternative reference to porosity in the OPA gouge. We like to keep the comparison to Orellana. However, we hope to have addressed the reviewers' issue in more detail by rewriting: We highlighted the difficulties in comparing these methods and refer to relevant literature:

*"…, it is unclear to which extend the applied methods are comparable to the BIB-SEM findings: are artificial openings excluded, are poro-elastic effects considered [cf. Sigal, 2009; Klaver et al., 2012; Hemes et al., 2013; Houben et al., 2014]?" (p.7, l.18)*

6.  Page 9, Line 1: Do you have other SiO2 modifications as quartz? You discuss this later, but should be mentioned before.

    We added:

    *"Moreover, trans-granular fracturing and boudinage is abundant in mica, calcite, feldspar, quartz (and/or $SiO_2$ grains, see section below) across several scales (Figure 9)." (p.9, l.1)*

7.  Page 10, Line 25: You may add: "… suggesting intracrystalline plasticity or fracturing parallel (001)."

    We agree and adapted the sentence (p.10, l.25).

8.  You may combine chapter 3.3.1 and 3.3.4. These two processes are somehow connected. You produce smaller grain sizes by cataclasis, which allow in the following frictional flow. This would be easier to read.

    We agree and combined both sections (now section 3.3.1 Cataclasis, abrasion and frictional granular flow).

9.  Page 12, Line 10ff: The argument using the Sr-isotope data is not understandable. I would always expect different Sr-data in veins, water and protolith caused by fractionation between different minerals (i.e. calcite versus clay) and the main difference between protolith and vein is the mineralogy. You may explain better, what is the argument (or omit this argument).

    We agree that our argumentation needs to be changed. The geochemical studies cited compared isotope ratios of pore waters and of calcite leachates from microtectonic structures and undeformed host rock. Moreover, they accomplished quantitative element analysis including REE. We reworded or argumentation into the following: "*This assumption is supported by geochemical studies (element and isotopic analyses), stating a limited time of fluid flow in the Main Fault that did not markedly affected the host rock and its pore-water composition [Clauer et al., 2017; Mazurek and De Haller, 2017]." (p.12, l.10ff).*

10. Page 13, Line 12ff: Many studies indicate that amorphous SiO2 is not stable over geological times above a certain temperature. Many arguments have been found, that amorphous SiO2 are precipitated and reorganized into quartz in geological time scale. Your findings, may discussed in this context: (1) is the amorphous SiO2 geological developed?; (2) what are T-t conditions to stay amorphous?

    We like this proposed course of action. For (1) we showed that amorphous $SiO_2$ is unlikely a preparation artifact, hence it is geological developed (p.10, l.12). We now cite a study that showed a minor effect of ion beam sample preparation on quartz crystallinity (Fu et al. 2005).

For (2) it is more difficult to us: even we firmly know the T-t conditions of the faulting from other studies (mainly Mazurek et al. 2011; <85°C, faulting initiated in late Miocene), we are unable to state if amorphous $SiO_2$ was stable at these conditions. Several studies examined amorphous $SiO_2$ in large-offset faults (as cited, p.10, l.19), but none yielded an explanation that we consider plausible for the setting of the Main Fault. Certainly, more work is needed here. We can only present our findings of amorphous $SiO_2$ without a valid explaining theory.

11. A personal comment:
The paper has many abbreviations (e.g., BIB_SEM, SAED, OPA, XRD, TOC, VR, DIA, GSD, etc.), which is sometimes difficult to read. This would be even more difficult, for a reader, which is not from the same scientific community.
We agree with the reviewer and reworded some sentences. Now, the abbreviations TOC, VR, DIA, SAED, XRD, FIB and OM are omitted. On the other hand, we considered that BIB-SEM, TEM, EDX and OPA are often used terms in our manuscript and thus better to be abbreviated.

**Referee comment by Jafar Hadizadeh**

1. The proposed hypotheses: The authors arguments regarding the "smear" hypothesis are vague at best. For example, why is it necessary for the possible source of the smear to be calcite-free (p.11, l.17)? Do the authors suggest the "smear" hypothesis as an alternative explanation only for the existence of the dark, calcite-free gouge type? What would be some relevant and expected characteristics of such "smeared-in" gouge? This hypothesis may even be considered untestable if a source for the smear cannot be identified, or is so highly speculative. The "authigenic generation" and/or "reworking of OPA" hypothesis on the other hand, is supported by evidence of progressive deformation (sharp difference in calcite content between the two gouge types; continuously traceable reduction in Riedel shear angles with rescept to shear zone borders, going from the calcite-rich gouge to the dark gouge; low clast-matrix ratios indicating higher strains in the dark gouge). As the authors have noted, different microstructural domains in the gouge indicate a number of deformation mechanism transitions that lead to relatively low-friction localization zones. This alone, questions the possibility of a "smearing" process since under steady P-T conditions smearing implies low friction of the smeared material. I agree with the authors that pure shear has been a factor in late development of the gouge layer geometries both due to change in deformation mechanism toward a less dilatant behavior as well as volume change. Such processes are more likely to be in-situ (or "authigenic" as the authors put it) as opposed to resulting from microstructural evolution of a "smeared" gouge. The evidence of amorphous SiO2, which is clearly presented in Figs. 14 and 16 of the manuscript, is consistent with late stage hydrothermal SiO2 in the gouge in agreement with comments by the anonymous reviewer regarding the stability of amorphous silica with T and age of the gouge. I suggest that the "smearing" origin of the studied gouge, as a distinct hypothesis, be removed. Given the authors current information, one may only include the possibility of spatially limited smearing event(s) within framework of the reworking hypothesis.

   We are happy to read that our favored hypothesis of authigenic gouge generation is agreed by the reviewer. Encouraged by this, we strengthen our argumentation that authigenic gouge generation is our favored hypothesis (p.11, l.24). However, we have not observed any "transition stage" or "birth place" of the gouge. Hence, we still have to stress that there is an alternative potential gouge material source: several, small, darker strands of clay found in undeformed rock from the study area (Figure 18) that could have been smeared. Of course, and likely, both hypothesized process have acted, with gouge being authigenic generated and subsequent smeared (cf. gouge 'intrusions', Figure 5). We just cannot surely tell. And as both mechanisms might have acted, all indicators for authigenic gouge are valid for a smear-in -and later reworked- hypothesis, too.

   We hope that our rewording and this explanation are convincing.

2. To strengthen the reworking hypothesis the authors need to strengthen the arguments that support OPA to scaly-clay transitional stage during progressive deformation. For this, they may need to carefully reexamine Figs 2, 4, 6 for microstructural relationships between scaly wall rock (OPA) and the rest of the gouge types and chemical map of calcite related to these boundaries. The section 4.2.1 describing the transition from OPA to scaly clay appears to me inadequate in view of its significance as mentioned above.

   For undeformed OPA to scaly clay, we do see transition structures indeed: The scaly clay aggregates come in varying 'scaliness', i.e. varying shear zone densities. We refer to our scaly-clay paper that examined this in detail (also SolidEarth). For the transition of scaly clay to gouge, it is more difficult: The gouge boundaries are always very sharp, even at SEM-scale –

there is no trend in porosity or Ca distribution or particle size or shape, but just a very sharp boundary. However, we proposed that the sigmoidal clast shapes in gouge type 1 (which also show a sharp boundary) suggest a progressive rework of the host rock (p.7, l.23; p.15, l.9; Figure 7). We added this now also to an earlier section (beginning of section 4.2.2 stage 2 – gouge generation, p.12, l.17).

3. In agreement with comments by anonymous reviewer about comparison of porosity measurements obtained via different methods, the use of mercury porosimetry in particular (cited estimates by the authors) is controversial because of uncertainties in non-fracture porosity values (usually overestimations) caused by elastic deformation of pores of different size. Estimates via careful image analyses is more reliable than this method. However, the authors have left readers to eyeball an estimate of the gouge porosity by looking at very few images in the manuscript (p.6, l.11ff), which could vary depending on the reader's experience with microstructural porosity. From Fig. 13e it is possible to suggest that higher porosity of OPA is due to primary porosity of mineral fragments (e.g. calcite and quartz) in the clay matrix.
As with the comment by the anonymous reviewer, we agree about the difficulty in comparing the methods. We believe to have this better addressed now, also by referring to papers on this matter. Moreover, we explain the porosity differences in Figure 13 in more detail now (p.7, l.15ff):
*"…Gouge material in between larger fractures shows clearly less visible porosity than undeformed OPA at the same resolution (Figure 13). Pores abundant in calcite fossils and in the less intense foliated clay matrix of undeformed OPA are completely absent in the gouge micrographs (Figure 13e vs. b).*

4. I strongly agree with the anonymous reviewer about combining sections 3.3.4 (frictional granular flow) and 3.3.1(cataclasis and abrasion).
We agree and combined both sections (p.8, l.28ff).

5. In Fig. 4B, traces of internal shears seem to have a light green color. If this is not a compositional color coding, it is better represented in thin black lines.
We changed the lines to become black (Fig.4). We supply the original Figures as vector files (pdf), so with the final paper version, you should be able to zoom-in to all mappings.

6. Fig. 7 lacks description for part D.
Right. We added: *"…(c) shaded light photograph and (d) sketch of sample V04 showing foliation…"* (Fig. 7).

7. Fig. 17e- where you have "(often with calcite clasts)", do you mean: common in calcite clasts?
Yes, we mean "common in calcite clasts". We changed it now. Thanks! (Fig. 17).